# Fabrication of glipizide loaded polymeric microparticles; *in-vitro* and *in-vivo* evaluation

Qaiser Rasheed[1], Kamran Ahmad Khan[1]*, Ghulam Razaque[2], Ashfaq Ahmad[3], Asif Nawaz[1], Naheed Akhtar[4], Kifayat Ullah Shah[1], Zahid Rasul Niazi[1], Muhammad Danish Saeed[1], Anila Alam[5]

1 Gomal Centre of Pharmaceutical Sciences, Faculty of Pharmacy, Gomal University, Dera Ismail Khan, Pakistan, 2 Faculty of Pharmacy, University of Balochistan, Quetta, Pakistan, 3 Riphah Institute of Pharmaceutical Sciences, Riphah International University, Gulberg Green Campus, Quetta, Islamabad, Pakistan, 4 Faculty of Pharmacy, University of Poonch, Rawalakot, Pakistan, 5 Faculty of Pharmacy, Sardar Bahadur Khan Women University Quetta, Quetta, Pakistan

* qaiserrasheedgsk@yahoo.com

**Data Availability Statement:** All relevant data are within the manuscript.

**Funding:** The author(s) received no specific funding for this work.

## Abstract

Controlled-release microparticles offer a promising avenue for enhancing patient compliance and minimizing dosage frequency. In this study, we aimed to design controlled-release microparticles of Glipizide utilizing Eudragit S100 and Methocel K 100 M polymers as controlling agents. The microparticles were fabricated through a simple solvent evaporation method, employing various drug-to-polymer ratios to formulate different controlled-release batches labeled as F1 to F5. Evaluation of the microparticles encompassed a range of parameters including flow properties, particle size, morphology, percentage yield, entrapment efficiencies, percent drug loading, and dissolution studies. Additionally, various kinetic models were employed to elucidate the drug release mechanism. Furthermore, difference and similarity factors were utilized to compare the dissolution profiles of the tested formulations with a reference formulation. The compressibility index and angle of repose indicated favorable flow properties of the prepared microparticles, with values falling within the range of 8 to 10 and 25 to 29, respectively. The particle size distribution of the microparticles ranged from 95.3 to 126 μm. Encouragingly, the microparticles exhibited high percent yield (ranging from 66 to 77%), entrapment efficiency (80 to 96%), and percent drug loading (46 to 54%). All formulated batches demonstrated controlled drug release profiles extending up to 12 hours, with glipizide release following an anomalous non-Fickian diffusion pattern. However, the drug release profiles of the reference formulation and various polymeric microparticles did not meet the acceptable limits of difference and similarity factors. In-vivo studies revealed sustained hypoglycemic effects over a 12-hour period, indicating the efficacy of the controlled-release microparticles. Overall, our findings suggest the successful utilization of polymeric materials in designing controlled-release microparticles, thereby reducing dosage frequency and potentially improving patient compliance.

**Competing interests:** The authors have declared that no competing interests exist.

## 1. Introduction

Diabetes mellitus (DM), a chronic metabolic disorder characterized by persistent hyperglycemia resulting from inadequate insulin production, resistance to insulin's peripheral effects, or both, poses a significant health challenge globally [1]. Projections by the International Diabetes Federation estimate that by 2022, 31% of individuals in Pakistan will be afflicted with diabetes, translating to nearly 33 million cases [2]. This alarming figure continues to escalate annually, with projections indicating an additional 200 million individuals expected to be diagnosed with DM by 2045, significantly impacting global public health [3].

Among the myriad complications associated with diabetes, microvascular (retinopathy, nephropathy, neuropathy) and macrovascular complications pose substantial risks, elevating the likelihood of cardiovascular disease by 2- to 4-fold [4]. Chronic hyperglycemia, coupled with metabolic abnormalities, predisposes patients with diabetes to organ damage across various systems.

Conventional medications used to manage hyperglycemia encompass sulfonylureas, biguanides, peroxisome proliferator-activated receptor (PPAR) agonists, and glucosidase inhibitors. Sulfonylureas stimulate insulin release from pancreatic islets, while biguanides suppress hepatic glucose production. These pharmacological agents are often employed either individually or in combination with other anti-diabetic drugs. However, their utility is hindered by significant drawbacks, including the propensity for severe hypoglycemia, weight gain, suboptimal therapeutic efficacy due to inappropriate dosing regimens, limited potency, altered side effects stemming from drug metabolism, lack of target specificity, and challenges related to solubility and permeability [5].

Despite the development of promising anti-hyperglycemic medications, the effective management of diabetes remains a considerable challenge. Enhancing current therapeutic approaches to ensure optimal glycemic control and mitigate long-term complications associated with diabetes represents a paramount objective [5]. Within this context, glipizide, a second-generation sulfonylurea, emerges as a focal point of the present investigation [6]. Analytical assessments indicate that glipizide, in its dried form, contains a minimum of 98% and a maximum equivalent to 102% of I-cyclohexyl-3-4-((2-(5-methylpyrazine-2-carboxamido) ethyl) benzenesulfonyl urea [7].

Glipizide, characterized by a pKa of 5.9, exists as a white, odorless powder. It exhibits solubility in dimethylformamide but is insoluble in water and alcohol [7]. Additionally, it dissolves in 0.1N sodium hydroxide. Despite being widely employed for the treatment of type II diabetes, the precise mechanism of action of glipizide remains unclear. Nonetheless, both in healthy individuals and those with diabetes, its primary mode of action appears to involve an augmentation of insulin secretion from the islets of Langerhans and a reduction in the metabolic clearance rate of insulin. Other proposed mechanisms include an increase in the number of insulin receptors, enhanced insulin binding to its receptors, and heightened sensitivity of peripheral tissues to insulin. Oral administration of glipizide results in almost complete absorption; however, its absorption is delayed when ingested with food. Therefore, for optimal efficacy, glipizide is best taken at least 30 minutes before meals. The recommended initial dose is 5 mg once or twice daily, with a maximum daily dose of 40 mg. Following oral ingestion, peak blood levels typically occur within 2 to 4 hours, with a half-life ranging from 3 to 7 hours and an average duration of action of 10 hours in healthy individuals.

Eudragit polymers, versatile polyacrylate polymers with varying degrees of solubility, are well-suited for formulating prolonged-release formulations [8]. Eudragit S 100 polymers can be utilized to create formulations that facilitate controlled release over predetermined time intervals, with release profiles tailored to specific requirements. To enhance therapeutic

efficacy and patient adherence, drug delivery can be optimized throughout the gastrointestinal tract (GIT). Celluloses such as hydroxypropyl methylcellulose (HPMC) find extensive pharmaceutical applications as binders, suspending agents, thickening agents, viscosity-increasing agents, and film-coating materials [9].

Microspheres, minute spherical microparticles serving as carriers for medications, offer numerous advantages in drug delivery. These microparticles, ranging in size from 1 μm to 1000 μm, can encapsulate, dissolve, bind, or entrap drugs within a polymer matrix [9]. They facilitate precise control over drug release, provide protection against degradation, improve stability, and enhance bioavailability. Microspheres find diverse applications including taste and odor masking, controlled-release formulations, stability enhancement, dosage reduction, and targeted drug delivery [10]. When selecting polymers for microsphere formulation, considerations include solubility, stability profile, safety, and economic feasibility. Compared to liposomes, microspheres exhibit greater stability in biological environments and can be surface-modified for targeted drug delivery. They are commonly utilized for regulated and sustained drug release, and find application in the treatment of various conditions such as ocular diseases, cancer, cardiovascular disorders, and inflammation [11].

The current study aimed to utilize polymers to formulate controlled-release microparticles for the oral delivery of glipizide. Various polymeric microparticles loaded with drugs have been successfully prepared using the solvent evaporation method [12, 13].

## 2. Materials and methods

### 2.1. Materials

Glipizide was generously provided by Wilshire Pharmaceutical, Pakistan, while Methocel K 100M was sourced from BDH Chemical Limited, Pool, England, and Eudragit S 100 was obtained from Rohm GmbH & Co. KG, Darmstadt, Germany. Sodium hydroxide and monobasic potassium phosphate were acquired from Aldrich Sigma, USA. The ingredients were utilized without further purification prior to use.

### 2.2. Glipizide standard calibration curve

A standard curve for glipizide was constructed in 0.1N hydrochloric acid (HCl) solution and phosphate buffer at pH 6.8. To prepare 0.1N HCl, approximately 8.8 mL of HCl (35%) was diluted to 1000 mL with distilled water. Subsequently, a drug stock solution (1 mg/mL) was prepared by dissolving 100 mg of glipizide in 100 mL of 0.1 N hydrochloric acid solution in a volumetric flask. A 10 mL aliquot from the stock solution was then withdrawn, transferred to a 100 mL volumetric flask, and diluted to 100 mL using 0.1 N HCl to achieve an initial drug dilution of 0.5 mg/mL. Similarly, four additional dilutions were prepared: 0.25, 0.125, 0.0625, and 0.03125 mg/mL. The absorbance of these dilutions was measured spectrophotometrically at 276 nm [14]. Subsequently, absorbance values were plotted against each concentration to generate a standard curve for glipizide. The same methodology was employed to construct the glipizide standard calibration curve in phosphate buffer pH 6.8.

### 2.3. Preparation and optimization

The creation of glipizide microparticles employs a straightforward solvent evaporation technique, as described by Madhusudhan et al. (2010). Various drug-to-polymer ratios were employed to formulate the microparticle formulations while ensuring a consistent drug concentration, as delineated in Table 1. The conditions of the method were meticulously optimized. In the initial phase, 30 mL of methanol was agitated at 1000 rpm for 10 minutes to

**Table 1. Composition of glipizide microparticles formulations.**

| Formulations | D:P | Glipizide (gm) | Eudragit S100+Methocel K100M (1:1, gm) | Methanol (mL) | Tegu 450 (mg) |
|---|---|---|---|---|---|
| F1 | 1:0.5 | 1.0 | 0.5 | 30 | 10 |
| F2 | 1:1 | 1.0 | 1.0 | 30 | 10 |
| F3 | 1:1.5 | 1.0 | 1.5 | 30 | 10 |
| F4 | 1:2.0 | 1.0 | 2 | 30 | 10 |
| F5 | 1:2.5 | 1.0 | 2.5 | 30 | 10 |

dissolve the polymers Eudragit S100 and Methocel K100M in a 1:1 ratio. Subsequently, in the second stage, a novel surfactant, Tegu$^{\circledR}$ 450, was introduced and mixed for 15 minutes using a magnetic stirrer. Following this, in the third phase, the medication was added and magnetically stirred for 20 minutes at 3000 rpm to ensure thorough incorporation. In the subsequent fourth step, the organic solvent was completely evaporated by allowing it to evaporate at room temperature for 24 hours. The resultant mixture underwent sieving using a No. 120 sieve in the fifth phase, and the resulting microparticles were collected thereafter. Finally, in the sixth and final stage, the microparticles were packaged in an airtight container and stored in a desiccator until further use.

## 2.4. Characterization

A standard approach was followed to characterize the microparticles, including DSC, FTIR, SEM, flow characteristics, percent drug loading, percent drug entrapment efficiency, drug content, and *in-vitro* drug release.

**2.4.1. Differential scanning calorimetry (DSC).** To evaluate the compatibility of the medication and the excipients, a DSC analysis of glipizide and the formulations was conducted. Briefly, 10˚C/min of heating was applied to 3 mg of each sample in a DSC heating pan (PerkinElmer, Pyres 6.0 DSC, and Waltham, MA, USA) to temperatures between 40 and 300˚C. For exothermic and/or an endothermic changes, DSC thermograms of each sample were obtained [15].

**2.4.2. Fourier Transform Infrared Spectroscopy (FTIR).** For the purpose of identifying possible chemical interactions between excipients or between medication and polymer in microparticle formulations, FTIR analysis [17] was carried out. ATR-FTIR was used to obtain the spectra of glipizide (API) and a few other formulations using a Spectrum Two FT-IR Spectrometer from Perkin Elmer in Waltham, Massachusetts, USA.

**2.4.3. Determining morphology.** Scanning electron microscopy (Carl Zeiss Inc., Oberkochen, Germany) was applied to find out the morphology of the microparticles. Samples of microparticles were carefully adhered on aluminum stubs with double-sided tape. The surface of the stub was left with a thin coating of particles after the stub was tapped to remove extra debris. At particular magnifications, representative parts were captured on camera. Before microscopy, samples of microparticles were coated with platinum using a Fine Auto Coater (JEOL, JEC-3000FC, and Bangkok, Thailand) [16].

**2.4.4. Determination of particle size.** The particle size of the microparticles was measured by employing a Master Sizer [18]. After 5 minutes of properly adding 2gram samples of microparticles to the instrument's sample fluid intake, the particle dimensions of the formulations had been obtained.

**2.4.5. Flow characteristics.** The flow characteristics of powder formulations are influenced by a variety of variables; including temperature, water content, chemical composition,

particle shape, and particle size distribution. The main factors used to assess particle flow are bulk density, tapped density, compressibility, and Hausner's ratio [17].

*2.4.5.1. Bulk density.* A 100 mL graduated cylinder was filled with a pre-weighed particle sample, and after giving it a light tap once, its volume was recorded. To determine the bulk density of the formulations, bulk volume values were introduced into the Eq [18].

$$\text{Bulk density} = \text{powder mass/packaging volume} \tag{Eq 1}$$

*2.4.5.2. Tap density.* The tap density was calculated in accordance with protocol. A 100 mL graduated cylinder containing pre-weighed microparticles was filled, and the cylinder was tapped for 30 minutes to achieve a consistent volume (tap volume). Eq 2's tap volume was used to get the tap density.

$$\text{Tapped density} = \text{mass of powder/tapped volume of pack} \tag{Eq 2}$$

*2.4.5.3. Compression index (CI).* The compression index [Behera et al., 2008] of the microparticles was calculated from the tap density and bulk density using the following Eq 3.

$$\text{Compressibility} = (\text{TappedDensity} - \text{BulkDensity/TappedDensity}) \times 100 \tag{Eq 3}$$

*2.4.5.4. Percent yield.* Percent yields [19] of microparticles were calculated for each batch. Batch considering the final weight of the product after vacuum drying compared to the total weight of the drug and polymers used in the preparation. It is calculated according to the Eq 4.

$$\% \text{ yield} = \text{microparticles obtained(mg)/initial amount of drug and polymer(mg)} \\ \times 100 \tag{Eq 4}$$

## 2.5. Drug entrapment and drug loading

Equal amounts of 30 mg of the medication were taken from each batch of microparticles, put in a 100 mL volumetric flask with 0.1N HCl, and well mixed for 30 min with a magnetic stirrer. At a measurement wavelength of 276 nm, the samples were filtered and subjected to spectrophotometric examination. Concentrations were obtained using the glipizide standard curve, and an equation was used to quantify the % drug loading and the effectiveness of drug entrapment [20].

$$\text{Entrapment Efficiency} = (\text{Actual Potency/Theoretical Potency}) \times 100 \tag{Eq 5}$$

$$\% \text{ drug loading} = \text{drug weight/microparticles weight} \times 100 \tag{Eq 6}$$

## 2.6. Dissolution study

The in vitro release of glipizide from microparticles was accomplished using the dialysis membrane technique [16]. As a separation media (inner dissolution medium), microparticles (5 mg glipizide) were added to an active dialysis membrane (MWCO 12000) holding 5 mL of 0.1NHCl solution. 350 mL of 0.1NHCl solution are contained in a bigger container once the dialysis bag has been sealed. The dissolving media was agitated at 100 rpm using a dialysis bag containing microparticles. The temperature was thermostatically maintained at 37 0.5°C while the water bath was shaken. We collected 5.0 mL [21] samples at predetermined times. Syringe filters were used to filter the samples, and they were then subjected to spectrophotometric

analysis at a measurement wavelength of 267 nm. Reference was handled in a similar manner as controls. For full drug extraction from microparticles, the sample was sonicated at least three times for 30 seconds at intervals of two minutes. Plotting was done between the cumulative percentage of drug release and time of drug release. Different calibrations have been applied to in-vitro release data. For the first two hours, 0.1N HCl solution was used as the dissolution medium, and for the following ten hours, phosphate buffer pH 6.8 was used in its place.

## 2.7. Kinetic models

To ascertain the drug release mechanism, the following models were used: Power Law, Zero-order, Ist-order, Highuchi, and Hixon-Crowell's [22]. MS Excel software was used to fit in-vitro drug release data from microparticles to the model. These models are given below:

Zero-order (Kinetics Model) [23]

$$W = K_1 t \tag{Eq 7}$$

K1 is rate constant and unit is concentration/time while t shows the time.

Ist- order (Kinetics Model) [24]

$$\ln(100 - W) = \ln 100 - K_2 t \tag{Eq 8}$$

K2 shows the Ist-order constant and Ln 100 indicates the initial concentration.

Hixon Crowel's (Erosion Model) [25]

$$(100 - W)^{1/3} = 100^{1/3} - K_3 t \tag{Eq 9}$$

Where, $(100 - W)^{1/3}$ is the initial concentration while, $100^{1/3}$ is concentration at time t and K3 is Hixon Crowel's erosion model constant.

Higuchi's (Diffusion Model) [22]

$$W = K_4 t^{1/2} \tag{Eq 10}$$

K4 indicates the design variable of the system.

Power Law

$$M_t / M_\infty = K_5 t^n \tag{Eq 11}$$

Mt / M denote the quantity of medication released at time t, while K5 shows the rate constant. It shows quasi-fickian diffusion at n = 0.5, anomalous non-fickian drug release kinetics at n = >0.5, and non-fickian ideal zero order kinetics at n = 1. n is the release exponent [23].

## 2.8. Comparison of dissolution patterns

For the purpose of comparing the profile of dissolution of the reference (API) and tested microparticles, difference factors (f1) and similarity factors (f2) were used to identify similarities as well as differences [26]. Since there was no controlled-release formulation on the pharmaceutical market, the drug (API) was used as a reference.

Difference Factor (f1)

$$f_1 = \left\{ \left[ \sum t = 1^n (R_t - Rt) \right] / \left[ \sum_{t=1}^{n} Rt \right] \right\} \times 100 \tag{Eq 12}$$

Similarity Factor (f2)

$$f_2 = 50 \times \log \left\{ [1 + (1/n) \sum\nolimits_{t=1}^{n} (R_t - Tt)^2]^{-0.5} \times 100 \right\} \qquad \text{(Eq 13)}$$

Where the number of pull points is indicated by n. Wt displays a possible weight element, and Rt is the drug release profile of the reference at time t. Tt is the profile of the microparticles.

## 2.9. Stability studies

Microparticles (F5) were examined in a stability chamber with conditions of 45˚C and 75% relative humidity. Drug release was then assessed six months after being exposed to settings with a higher temperature.

## 2.10. *In-vivo* studies

Healthy local breed rabbits were selected and used for the *in-vivo* investigation. The *in-vivo* study protocols were approved by the university ethical committee via letter number 205/QEC/GU, Gomal University, Pakistan. For the in vivo testing in this work, optimized Glipizide microparticles (F5) and reference (API) were used. A rabbit typically weighed 1.5–2.0 kg. Eight of the sixteen rabbits were picked to be utilized for inducing diabetes, and after doing so in part of them, the survivors were divided into two groups, A and B. Eight more rabbits were allocated into groups C and D as well. Pharmacokinetic studies and hypoglycemia effects were explored.

**2.10.1. Plasma sugar level before diabetes induction.** Prior to the onset of diabetes, the plasma sugar level was evaluated. The rabbits were maintained on a 12-hour fast but were given free extra water throughout the trial. 0.5 mL of blood was taken from each rabbit's ear vein, collected in a little tube, and allowed to clot. The plasma was then transferred to other tubes and kept in freezers at -40˚C after 150μL of serum was centrifuged for 15 minutes at 2000 rpm in a subsequent 5 mL glass tube. The glucose level was tracked using a Glucometer for 12 hours.

**2.10.2. Induction of diabetes.** Alloxan monohydrate was given to the rabbits in line with the directions [27]. The purpose of the steel cage used to confine the rabbits was to collect blood. The drug was injected intravenously into the rabbit's marginal ear veins after, if required, dabbing xylene into that specific vein. The xylene helped make the ear veins visible and useful as injection sites. Each of the eight rabbits, weighing between 1.0 and 1.2 kilogram's, received 75 mg of alloxan monohydrate. After receiving the medication, it was discovered that 2 rabbits died while the others lived. The 75 mg/kg body weight suggested dosage was followed while administering the dose [28]. The lowest death rate of 20% that was produced by the dosing regimen that was shown to be the most efficient could not be further reduced. To prevent the predicted hypoglycemia brought on by the Alloxan injection, each rabbit received an oral dosage of 2 grams of glucose per kg of body weight combined with 10 ccs of distilled water [29]. The drug was diluted in 8 ccs of distilled water in a Petri plate and then injected into the marginal ear veins using an insulin needle attached to a 10 cc syringe. The remaining rabbits with an RBS of more than 200 mg/dl were classified as diabetic and used for further testing eight days after receiving Alloxan. Alloxan doses much lower than those that had already been administered were repeated at intervals of between 5 and 10 days if it turned out that the rabbits didn't have diabetes. For example, if a rabbit had previously taken 75 mg/kg, the next two doses would be 55 mg/kg and 30 mg/kg, presuming the rabbit had been cleared of diabetes. Each Alloxan monohydrate injection was preceded by a fresh assessment of the rabbit's weight and overall health. After receiving a third Alloxan dosage, the majority of the rabbits developed

diabetes, which ought not to have exceeded 200 mg/kg. Some had ultimate doses up to 240 mg/kg and required 4 trials. The data analysis revealed that two unusually resilient animals were able to resist extremely high cumulative doses of 280 mg/kg and as many as 5 trials without acquiring diabetes. Blood sugar levels in each of the diabetic rabbits who survived ranged from 180 mg/dl to 650 mg/dl. The rabbits had type-2 diabetes because they could survive without insulin injections, which were necessary for our animal diabetic model. After starting a diabetic treatment, the sugar level was measured using a glucometer.

**2.10.3. Administration of test and reference formulation.** Each of the two diabetes animal groups (A and B) contained three diabetic rabbits (n = 3). Group A got F5 (optimized microparticle formulations) at an 800 mg/kg dosage, whereas group B received glipizide (API) at an 800 mg/kg dose. Blood was drawn from the marginal ear vein on a regular basis at intervals of 0, 1, 2, 3, 4, 5, 6, 7, 8, 9, 10, 11, and 14 hours. Serum was quickly separated by centrifugation at 28a speed of 3000 rpm for 25 minutes after the rapid collection of blood samples in pre-heparinized Eppendorf tubes. The levels of serum glucose were determined using a glucometer. After the method's validation, a relative standard deviation of 1.04% was identified. The rate of fall in blood glucose levels over time was calculated for one rabbit (n = 3). To represent each outcome, the standard deviation was added to the mean. The statistical variances have been studied using one-way analysis of variance (ANOVA). Significant was defined as a 0.05 $p$-value [30].

**2.10.4. Pharmacokinetic parameters.** There were still four rabbits from Groups C and D, for a total of eight (n = 8) for the pharmacokinetic testing. This also made advantage of the parallel design. One group (Group C) got the reference formulation, whereas Group D received the reference (API) using syringes with the injection side chopped off by 2 cm so order to prevent any injury to the mouth. Regularly, 0.5 mL of blood from the ear's marginal vein was removed from each rabbit, kept in minuscule 3 mL tubes, and then allowed to clot. A second 3 mL glass tube with 200 mL of serum was then inserted into the first one. After centrifuging the plasma from this tube at 2800 rpm for 15 minutes, it was transferred to fresh tubes and stored at -20˚C. Glipizide was measured on the HPLC column using a syringe and 5mL of the reconstituted solution. In order to improve the accuracy of the approach, blank plasma was created and mixed with known glipizide concentrations to produce concentrations of 5.0, 10.0, 20.0, 40.0, 80.0, and 160ng/mL as well as a constant concentration of 160ng/mL. This HPLC device has an NCI 900 UV wavelength detector and a binary pump solvent delivery system with an integrator. Chromatographic detection was carried out throughout a distribution range of 5 m pore sizes using an ODS Hypersil C18 stainless steel analytical column with a refillable pre-column. The detector operated at a wavelength of 276 nm. In a ratio of 44:24:24 v/v, acetonitrile, methanol, and 75 mM phosphate buffer made up the mobile phase for the HPLC analysis. Peak values showed that following inspection at a flow rate of 1.0 mL/min, a substantial portion of plasma was still intact. The regression equation used to calculate the medication's plasma concentration took into account the peak elevation ratio of the active component (glipizide). The plasma samples were kept so that the next step may be utilized to eliminate the glipizide before administering the injection. A glass vial holding 100 mL of plasma sample, a Teflon screw top, 100 mL of 1 M NaOH, and 6 mL each of pentane dichloromethane in a 75:25 ratio were added while being continuously agitated. The created solution was centrifuged for a further ten minutes at a speed of 3000 rpm after 1–2 minutes of vortex mixing. After centrifugation separated the top organic layer, the solvent was evaporated at a constant nitrogen flow temperature of 40˚C until the extract was completely dry. The final residue was collected, reconstituted with 100 mL of acetonitrile, and then let to stand in a glass tube for one minute. The pharmacokinetic information for the treated rabbit, which comprised $C_{max}$ and $T_{max}$, half-life, AUCo, $AUC_{0-t}$ $AUC_{0-\infty}$, and $K_e$ values, was calculated using Software Kinetica.

Additionally computed were the apparent volume of distribution (Vd/f): dosage/(AUC0-x $K_e$) and the elimination half-life ($t_{1/2}$). The absorption profile of each individual was determined using the equation below.

$$\% \text{ absorbed} = Ct + K_e AUC_{o-t}/K_e \quad \text{(Eq 14)}$$

Where $K_e$ is rate constant of the elimination. The drug amount is denoted by Ct. AUC represent total drug quantity in rabbit's body.

**2.10.5. Methods of sacrifice of animals.** In accordance with standard ethical guidelines, no intentional sacrifice of animals was part of the study protocol. The study adhered to internationally accepted ethical principles for the care and use of animals in research. Any animal death that occurred during the study was an unintended and unfortunate event, likely related to the physiological effects of the substances administered, such as Alloxan monohydrate, which is known to induce significant physiological stress. At all times, the health and well-being of the animals were closely monitored by qualified personnel, and all efforts were made to prevent unnecessary harm. Additionally, the study was approved by the institutional Ethical Review Board (approval no. 205/QEC/GU), ensuring compliance with the necessary ethical standards.

**2.10.6. Methods of anesthesia and/or analgesia.** The study did not involve surgical or invasive procedures that would typically necessitate anesthesia. However, the administration of substances was conducted with utmost care to minimize any potential discomfort. Additionally, following ethical guidelines, if any animal had shown signs of distress beyond what was expected during the course of treatment, appropriate interventions would have been taken to mitigate discomfort, including analgesia or other supportive care measures as needed.

**2.10.7. Efforts to alleviate suffering.** Throughout the study, efforts were made to alleviate any potential suffering. For example, after the administration of Alloxan monohydrate to induce diabetes, animals were monitored for hypoglycemia, and oral glucose was administered to prevent severe hypoglycemic episodes. Moreover, the animals were under continuous supervision to ensure that any signs of undue distress were promptly addressed. The research team took all reasonable measures to safeguard the animals' well-being, consistent with ethical research standards.

## 2.11. Applicable statistics

Each of the analyses of the dissolution patterns were subjected to a one-way analysis of variance at the significance level (0.05). The remainders of the research results were handled as mean SD.

## 2.12. Ethics declarations

The *in-vivo* study protocols were approved by the university ethical committee via letter number 205/QEC/GU, Gomal University, Pakistan.

## 3. Results and discussion

### 3.1. Standard calibration curve

It was discovered that a glipizide standard calibration curve has an r2 of 0.9992, indicating linearity between concentration and absorbance. The total medication dosage was calculated using this. A standard calibration curve with an r2 value of 0.9999 was also built in phosphate buffer at pH 6.8. As shown in Figs 1 and 2.

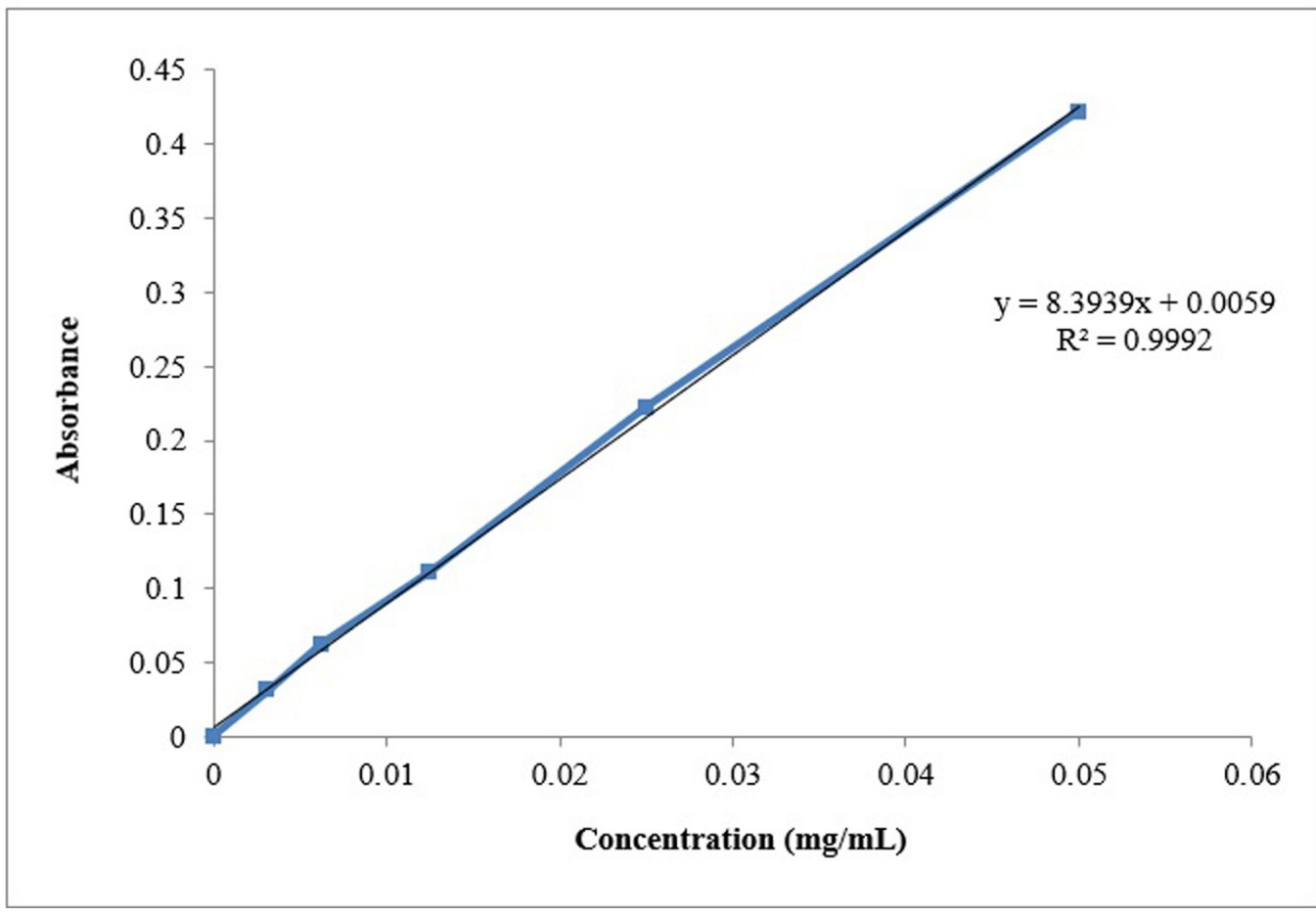

**Fig 1. Standard calibration curve of glipizide in 0.1N HCl.**

### 3.2. FTIR analysis

Standard techniques were followed while conducting FTIR measurements, and it was discovered that there was very little difference between the peaks of the pure medication and its formulation. At 2911.44 cm-1, the primary functional group CH & CH2 aliphatic stretching group first occurred. It was also present in all formulations F1 to F5. As a result, there was no evidence of a drug-excipient interaction in the microparticles. The main functional groups of the active drug are still present, demonstrating that the drug and polymer are compatible, according to the results of the author's [20] FTIR examination of the drug ingredient and its formulations As shown in Fig 3.

### 3.3. Differential scanning chromatography (DSC)

The DSC analysis was carried out to validate any potential drug-excipient interactions in the microparticles, however none were found as shown by the chromatograms since the formulations and the drug's (API) end points were both 210˚C. At 210˚C, a strong endothermic peak of the drug was noticed, signifying drug melting. The particular endothermic peak verifies that glipizide melts at this temperature. The exact peaks in drug-loaded microparticles that corresponded to the melting point of glipizide were readily discernible at 210˚C, demonstrating the compatibility of the polymer with the drug. The DSC results revealed that the formulation

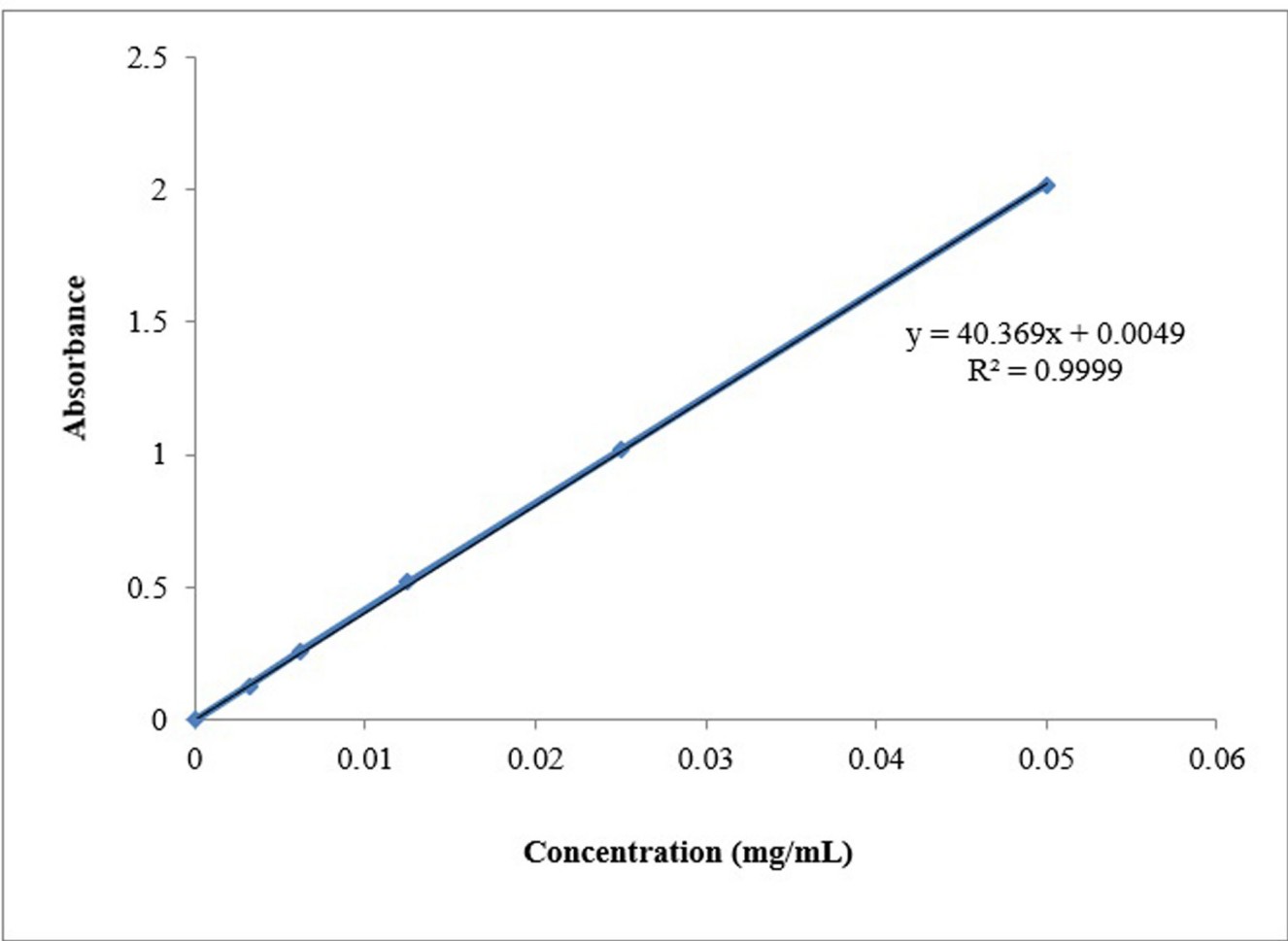

**Fig 2. Standard calibration curve in phosphate buffer pH 6.8.**

conditions and process factors had no negative effects on medication stability. The current findings are in line with the findings of the author, who conducted a DSC study of drug and polymer formulations and discovered no interactions between the excipients in the formulation and the drug As shown in Fig 4.

### 3.4. Flow properties

After determining the flow characteristics of several formulations, it was noticed that all parameters were found to be within reasonable bounds, as shown in Table 2. The findings revealed that the angle of repose of different formulations ranged from 25.31° to 29.54°, which was determined to be within USP limitations [17] for angle of repose (25° to 30°), demonstrating efficient particle flow. The compressibility indices of the formulations varied from 8.9 ±1.248 to 10.0 ±0.016, which was also adequate for a good flow of particles (0 to 10%; USP) [17].

### 3.5. Analysis of particle size and morphology

A formulation size analyzer was used to do particle size analysis. The diameters of the particles were discovered to range from 95.3±0.139 μm to 128±0.009 μm. As the amount of polymer in

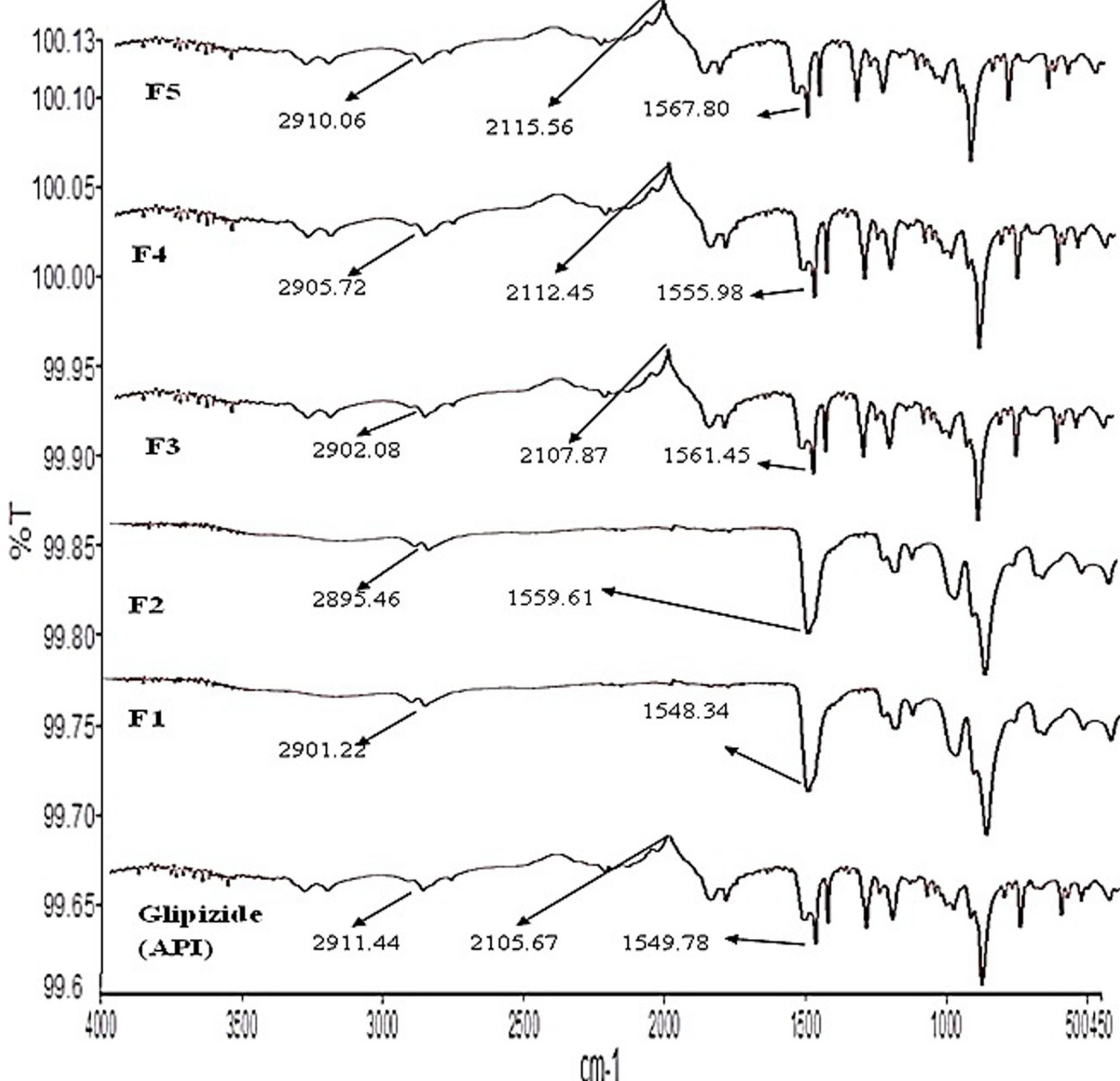

**Fig 3. FTIR spectra with functional groups of glipizide (API), F1 to F5.**

various formulations was gradually increased, the size of the microparticles increased as more drugs were trapped inside of them. These findings are congruent with the findings of the current investigation. When other researchers [16] studied the microparticles' particle sizes, they found that the size of the particles rose (2 m to 32 m) as the amount of polymer increased. The findings of the SEM examination are shown in Fig 4, and it was discovered that the surfaces of the microparticles were rough. The image of the liposperes formulations was examined using SEM analysis to determine the effect of surfactant concentration, SS, and BA on the shape of the liposperes [31]. Due to the OL's greater T-80 concentrations and lower polymer (2.0%)

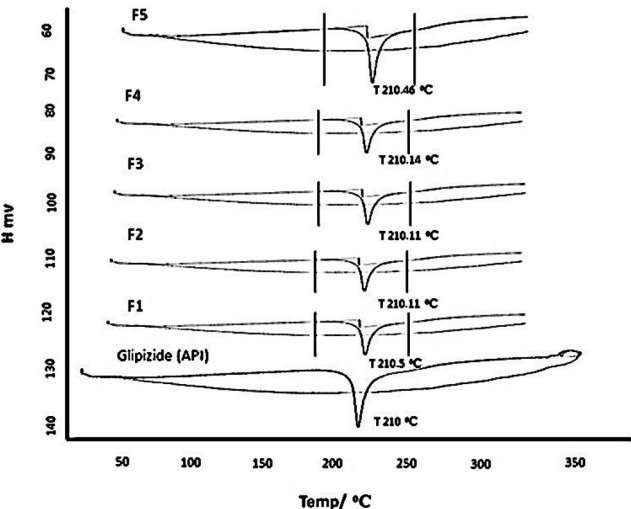

**Fig 4. DSC chromatogram of glipizide (API) and various formulations.**

quantity, the SEM results demonstrated that the lipospheres formulation had a homogeneous and smooth surface. In the current work, surfactant was also discovered to have a crucial function in the stabilization of BA emulsion droplets as well as the formation of smooth micron-sized lipospheres. Additionally, the SEM image showed that a faster stirring rate had helped to produce spherical lipospheres and had stopped the formation of aggregates of hydrophobic lipid polymer. Table 3 and Fig 5 present the findings.

## 3.6. Entrapment efficiency

All microparticles' percent entrapment efficiency was calculated, and it was discovered that F1 (the reference formulation) had no entrapment efficiency. According to Table 4, the formulations' entrapment efficiencies varied from 80.56±0.452 to 93.32±0.183%. Formulation F5 had the highest encapsulation effectiveness (96.23±0.764%) and more pharmaceuticals were trapped. When the medication was more fully enclosed within the polymer, the authors [31] reported high encapsulation efficiency, which is consistent with the findings of our research.

## 3.7. Percentage/proportional yield

When microparticle recovery was calculated, the findings varied from 69.86±0.462 to 77.15 ±0.036%, with F5 recovery being higher than that of other microparticle formulations at 77.15 ±0.036 (Table 4). All microparticles were discovered to provide good yield percentages. Microsphere yields were good percentage-wise, according to the author [27].

**Table 2. Flow properties of glipizide microspheres.**

| Code | Bulk density (g/mL) | Tapped density (g/mL) | Angle of repose (o) | Compressibility Index (%) |
|---|---|---|---|---|
| F1 | 0.623±0.146 | 0.771±1.125 | 27.36± 3.389 | 8.9± 1.248 |
| F2 | 0.625±0.158 | 0.716±1.841 | 28.93±0.341 | 9.46± 1.017 |
| F3 | 0.711±1.083 | 0.931±2.543 | 25.31± 1.456 | 10.0±0.016 |
| F4 | 0.627±0.238 | 0.718±0.115 | 27.82± 1.270 | 9.47±0.783 |
| F5 | 0.70±1.422 | 0.777±0.255 | 29.54±2.820 | 9.09±0.232 |

Table 3. Particle size of microparticles of glipizide.

| Formulations | Particle size (µm) |
|---|---|
| F1 | 100.8±0.105 |
| F2 | 115±0.164 |
| F3 | 95.3±0.139 |
| F4 | 120±0.032 |
| F5 | 128±0.009 |
| F6 | 126±0.014 |

## 3.8. Percent drug loading

Drug loading of microparticles ranging from 46.67±0.005 to 54.81±0.167% was assessed as the amount of polymer rose, showing that drug loading also increased and the polymer exhibited good drug loading. F5 (54.81±0.156) demonstrated increased drug loading. The authors [32] noted that higher drug loading was seen when the quantity of polymer was increased in various formulations. Table 4 displays these outcomes.

## 3.9. *In-vitro* drug release study

The polymer extended the drug release rate for 12 hours, according to estimates of drug release rates for the microparticles (F1 to F5). In 2 hours, reference (API) released 100% of the drug, and in 12 hours, F1 released 95.86% of the drug. In a 12-hour period, F2 released 89.46% of the drug and F3 released 86.35%. At 12 hours, F4 and F5 released 84.34% and 82.28% of glipizide, respectively (Fig 3). Compared to the reference medication glipizide (API), which only extended the drug release by two hours, this polymer combination (Eudragit S 100 and Methocel K 100 M) increased the drug release rate by up to 12 hours. Compared to the other formulation, F5 demonstrated longer drug release rates. Eudragit S 100, a hydrophilic polymer that resists water absorption, and Methocel K 100M, a hydrophilic polymer that expands when in contact with medium, both have the potential to delay drug release from microparticles. This study's findings are comparable with those of another study that discovered polymer materials, such as Eudragit S 100, extended the rate of medication release. According to the authors of

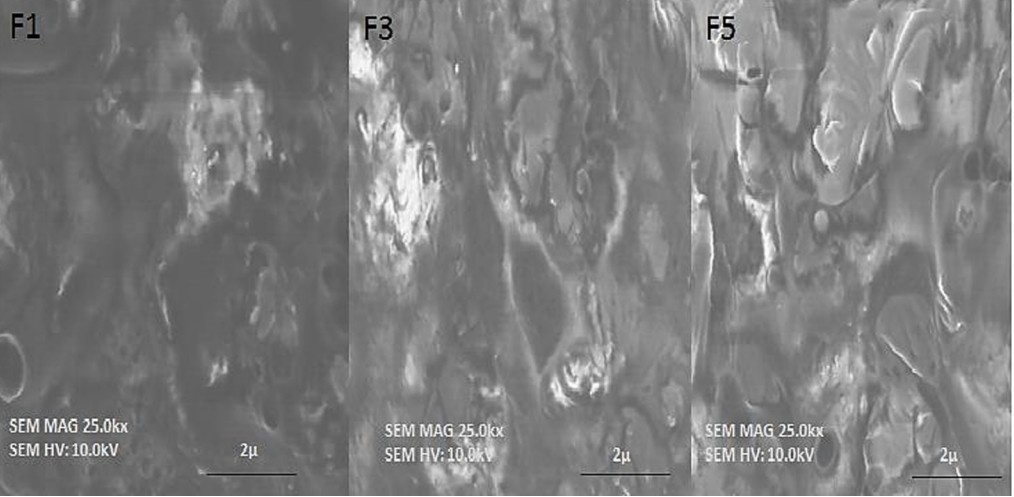

Fig 5. SEM images of F1, F3 and F4.

**Table 4. Results of % entrapment efficiency, % drug loading, and % yield.**

| Formulations | % Entrapment efficiency | % Dug loading | % Yield |
|:---:|:---:|:---:|:---:|
| F1 | 80.56±0.452 | 46.67 ± 0.005 | 66.54±1.452 |
| F2 | 88.87±0.853 | 47.81 ± 0.167 | 68.32±0.154 |
| F3 | 89.68±0.388 | 48.76 ± 0.014 | 68.53 ± 0.521 |
| F4 | 93.32±0.183 | 49.54 ± 0.034 | 69.86±0.462 |
| F5 | 96.23±0.764 | 54.81±0.156 | 77.15±0.036 |

the current investigation, the polymer Methocel K 100 M helped to prolong the release of clarithromycin and famotidine. Fig 6 displays these findings.

## 3.10. Drug release mechanism

To determine which kinetic model the drug release data adheres to, data were fitted into a variety of kinetic models. $R^2$ values for the first-order kinetic model varied from 0.3425 to 0.4336, indicating that the medication was not released according to first-order release kinetics. When data was fitted in the Higuchi model, the $r^2$ values ranged from 0.5016 to 0.6542 and showed that drug was released by pseudo diffusion mechanism and F5 followed more this model than rest of formulations. When data was fitted in the zero-order kinetic model, the $r^2$ values ranged from 0.5254 to 0.6891 that indicates that drug release data slightly followed this model. $R^2$ scores between 0.6026 and 0.6894 showed that the medication was released when the data were fitted into Hixon Crowell's model. F5 ore matched this approach more than other

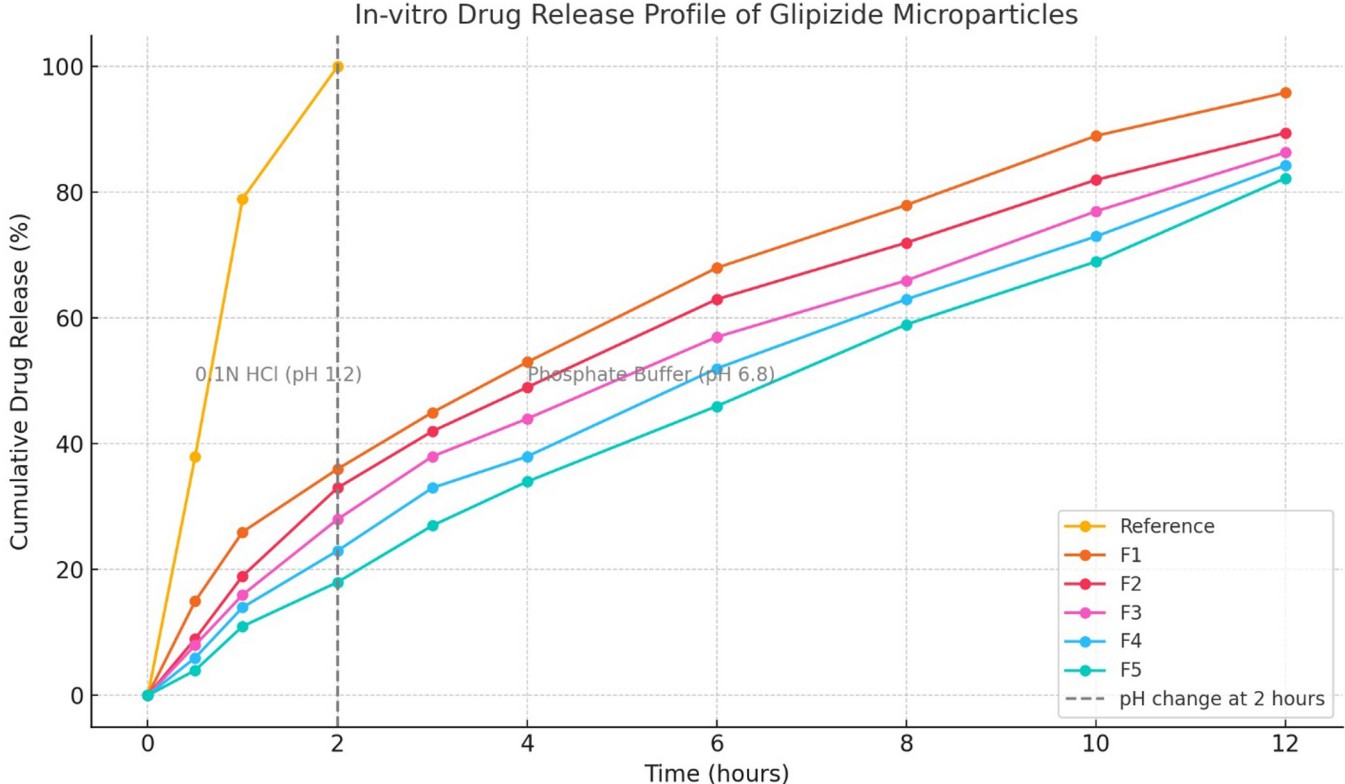

**Fig 6. In-vitro drug release profile of formulations.**

**Table 5. Drug release mechanisms.**

| Code | First- order kinetic model | Zero- order kinetic Model | Highuchi model | Hixon crowell's model | Power law | | Drug release mechanism |
|------|------|------|------|------|------|------|------|
| | R2 | R2 | R2 | R2 | R2 | N | |
| F1 | 0.4336 | 0. 3435 | 0.5467 | 0.6421 | 0.9532 | 0.6436 | ANFD |
| F2 | 0.4214 | 0.6353 | 0.5381 | 0.6321 | 0.9654 | 0.7851 | ANFD |
| F3 | 0.4265 | 0.6215 | 0.5016 | 0. 6026 | 0.9867 | 0.8453 | ANFD |
| F4 | 0.3425 | 0.5991 | 0.5217 | 0.6643 | 0. 9753 | 0. 6847 | ANFD |
| F5 | 0.3541 | 0.5254 | 0.6543 | 0. 6894 | 0.9896 | 0. 8548 | ANFD |
| Reference | 0.8456 | 0. 6891 | 0.0254 | 0.1849 | 0.1843 | 0.2642 | FD |

FD = fickian diffusion, ANFD = Anamolous non-fickian diffusion

equations. When the power law kinetics model was used, as shown in Table 5, the drug release kinetics was linear, with r2 values that varied from 0.9532 to 0.9896. These n-values are (0.6436–0.8548) from F2 to F6 suggesting that a non-capricious diffusion mechanism was used to release the medication. Contrarily, the reference formulation displayed Fickian diffusion and non-linear drug release kinetics [33]. The dynamics of the F5 formulation were almost zero-order. The findings of the present investigation are consistent with those of previous authors [34], who discovered that anomalous non-fickian processes were used to release losartan potassium.

## 3.11. Dissolution comparison

There was no agreement found between the dissolution profiles of the reference formulation and the test microparticles when comparing the dissolution profiles of the reference formulation and the test formulation using the difference and similarity factors. As can be seen in Table 6, the values of f1 (24.17 to 45.2) and f2 (15.04 to 27.68) were beyond the ranges that accommodated f2 (50–100) and f1 (1–15). These findings are supported by authors [35] who discovered that the formulation drug release rates and reference drug release rates did not agree. Table 6 presents these findings.

## 3.12. *In-vivo* hypoglycaemic effect and pharmacokinetic parameters

In-vivo testing of the microparticles of Eudragit S100, Methocel K100M, and Tegu-450 was done on healthy, normal rabbits by observing the hypoglycemic response brought on by the oral administration of pure Glipizide and controlled-release Glipizide microparticles at a dose equating to 800 g.kg$^{-1}$. Significant differences in the hypoglycemic effects of regular Glipizide and the chosen formulation were seen when the medication was delivered. The micro particles (F5) with a drug-to-polymer ratio of (1: 2.5) have maintained hypoglycemic effect for a period of 8 hours, whereas for reference (API) it was 1–4 hours due to the drug being released in a

**Table 6. Results of dissolution profile comparison.**

| Reference vs test formulation | f1 | f2 |
|------|------|------|
| reference vs F1 | 28 | 21 |
| reference vs F2 | 24.17 | 2768 |
| reference vs F3 | 38.24 | 18.06 |
| reference vs F4 | 41.63 | 16.55 |
| reference vs F5 | 45.209 | 15.04 |

**Table 7. Random blood sugar level before administration of alloxan, after alloxan administration, the reference and microparticles (F5).**

| Time (hours) | RBS level before Alloxan (mg/dl) | RBS level after giving Alloxan (mg/dl) | RBS (mg/dl) after giving Reference (API) | RBS (mg/dl) after giving microparticles (F5) |
|---|---|---|---|---|
| 0 | 119.18±1.32 | 452.43±0.17 | 216.96± 0.12 | **218.65 ± 1.32** |
| 1 | 107.09±0.22 | 495.14±0.12 | 205.92±0.18 | **207.94±1.08** |
| 2 | 123.55±0.14 | 491.08±0.11 | 171.12± 1.09 | **173.86±1.17** |
| 3 | 104.07±0.12 | 456.03±0.04 | 170.47 ± 0.05 | **171.67±0.03** |
| 4 | 138.27±0.13 | 428.14±0.16 | 134.08 ±0.34 | **136.72±0.05** |
| 5 | 142.45±0.6 | 432.15±0.34 | 120.45 ±0.22 | **124.23±0.84** |
| 6 | 123.55±0.03 | 431.13±0.15 | 111.76± 0.41 | **123.88±0.43** |
| 8 | 104.07±0.08 | 427.34±0.23 | ---- | **118.65± 0.54** |
| 10 | 138.27±0.12 | 429.15±0.06 | ---- | **114.34±0.95** |
| 12 | 142.45±0.14 | 431.13±0.04 | ---- | **111.38±0.54** |

RBS is random blood sugar

controlled manner. 20% reduction in glucose levels is considered significant for hypoglycemic effect (Table 7).

## 4.13. Pharmacokinetic parameters

In this study, we employed local breed animal models to take a look into how optimized formulation (F5) microparticles and reference (API) are performed *in vivo*. Drug level was observed for prolonged time in plasma, the longer $t_{1/2}$ half-life and time $T_{max}$ are essential. The area under the curve of F5 indicates that polymers significantly reduced the rate of drug absorption. It can be seen in Table 8. The Glipizide microparticles (F5) had a substantially different $T_{max}$ (1.51.15hrs) when contrasted with the reference (API) $T_{max}$ (5.30.5 hours), where the P-value was 0.05 [36]. This proved that polymers contributed to the medicine's half-life. The F5 had half-life ($t_{1/2}$) of 5.95±0.25 hrs, with a *p*-value of 0.05, when compared to the reference (API), which had a $t_{1/2}$ of 1.06±0.5 hrs. For F5, longer $T_{max}$ and half-life ($t_{1/2}$) values proposed a longer absorption phase and expanded drug presence in the body. The peak concentration ($C_{max}$) values for the F5 formulation were also found to be substantially higher (60.321.84ng/mL) than those for the active ingredient (620.35ng/mL), with a *p*-value of 0.0001. While the AUC0-∞, for the reference (API) was 1865±0.54 ng.hr/mL and 1926±0.13 ng.hr/mL for the formulation F5, the $AUC_{0-t}$ for the CR formulation F5 was likewise 2143±0.14 ng.hr/mL. It was found that for formulation F5, the total Clearance Cltotal for reference (API) and elimination rate constant $K_{el}$ were, respectively, 0.016±0.002 h$^{-1}$ and 0.083±0.021 hr/(ng/

**Table 8. Optimized microparticles (F5) and reference (API) formulation pharmacokinetic parameters.**

| Pharmacokinetic parameters | Reference (API) | F5 | One-way ANNOA (*p*-values) |
|---|---|---|---|
| $C_{max}$ (ng/mL) | 62.32 ±0.35 | 60.75 ± 1.84 | *p*<0.05 |
| $T_{max}$ (h) | 1.5 ±1.15 | 5.3 ±1.21 | *p*<0.05 |
| $T_{1/2}$ (hrs) | 1.06 ±0.52 | 5.95 ±0.42 | *p*<0.05 |
| $AUC_{0-t}$ (ng.hr/mL) | 1665 ±0.32 | 2143 ± 0.14 | *p*<0.05 |
| $V_{d/f}$ (L/kg) | 12.8 ±0.91 | 20.14 ±0.89 | *p*<0.05 |
| $AUC_{0-\infty}$ (ng.hr/mL) | 1865 ±0.54 | 1926 ± 0.13 | *p*<0.05 |
| $Cl_{Total}$ (L/hour) | 0.016 ±0.02 | 0.083±0.01 | *p*<0.05 |
| $K_{el}$ (h$^{-1}$) | 0.116 ±0.04 | 0.032±0.05 | *p*<0.05 |

**Table 9. Drug stability study data of F5 formulation.**

| S. No | Days | Drug release from F5 after exposure to accelerated temperatures 40 ± 0. 2˚C | % yield of F5 after exposure to accelerated temperatures 40 ±0. 2˚C |
|---|---|---|---|
| 1 | 0 | 81.11± 00 | 76.32±0.03 |
| 2 | 30 | 81.98±0.015 | 76.12±0.01 |
| 3 | 45 | 81.15±0.09 | 76.06±0.04 |
| 4 | 90 | 81.4±0.13 | 76.02±1.02 |

mL). Researchers [37, 38] have evaluated Glipizide microparticle formulations in vivo; however the data from these studies showed that the results were slightly varied in terms of pharmacokinetic constraints, which may be due to subject and formulation variances.

### 4.14. Study of stability

The F5 formulation underwent a stability trial by being kept at 40±0.2˚C and 75% RH for three months. At the scheduled intervals, the sample's drug content was examined. It was discovered that the drug content of the F5 formulation had not changed much. This shows that F5 was stable at the subsequent temperature (Table 9).

## 5. Conclusion

From the results of this project, it can be concluded that the prepared microparticles extended the drug release rate up to 12 hours and released the drug by non-diffusive diffusion. A statistically significant variation was observed between the dissolution patterns of the reference and the test microparticles. The *in-vivo* hypoglycemic and pharmacokinetic parameters were also determined. These microparticles can be effectively used as controlled-release dosage formulation for the treatment of diabetes. The polymer combination employed in the study can be effectively utilized to develop controlled-release formulations.

An important consideration in the development of sustained-release formulations is the safety of the excipients used. Both **Eudragit S100** and **Methocel K100M** are polymers that have been extensively utilized in oral drug delivery systems due to their proven safety profiles. **Eudragit S100** is a widely accepted polymer for controlled drug release and is FDA-approved for use in oral formulations. **Methocel K100M** (hydroxypropyl methylcellulose) is similarly approved and used as a release-retarding agent in various oral medications. Numerous studies have demonstrated that these polymers exhibit **minimal gastrointestinal absorption**, ensuring that they remain largely non-absorbable and are safely excreted from the body. Furthermore, they are associated with low toxicity, even in prolonged use, confirming their **biocompatibility** and **patient safety** in oral formulations. These characteristics, along with their ability to provide controlled drug release, make them ideal candidates for the development of sustained-release microparticles, as demonstrated in this study.

## Acknowledgments

The authors are grateful to Gomal University, Dera Ismail Khan, Pakistan for all possible facilities through Gomal Center of Pharmaceutical Sciences.

## Author Contributions

**Conceptualization:** Naheed Akhtar, Zahid Rasul Niazi, Muhammad Danish Saeed, Anila Alam.

**Data curation:** Ghulam Razaque.

**Formal analysis:** Ghulam Razaque, Ashfaq Ahmad.

**Investigation:** Ashfaq Ahmad.

**Methodology:** Qaiser Rasheed, Kamran Ahmad Khan, Asif Nawaz, Muhammad Danish Saeed.

**Project administration:** Asif Nawaz.

**Resources:** Naheed Akhtar, Kifayat Ullah Shah.

**Software:** Naheed Akhtar.

**Supervision:** Kamran Ahmad Khan, Zahid Rasul Niazi, Muhammad Danish Saeed.

**Validation:** Kifayat Ullah Shah.

**Writing – original draft:** Qaiser Rasheed, Anila Alam.

**Writing – review & editing:** Anila Alam.

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
