## [Decision Letter · Decision Letter 0]

6 Aug 2024

PONE-D-24-12544Fabrication of Glipizide Loaded Polymeric Microparticles; in-vitro and in-vivo EvaluationPLOS ONE

Dear Dr. Rasheed,

Thank you for submitting your manuscript to PLOS ONE. After careful consideration, we feel that it has merit but does not fully meet PLOS ONE’s publication criteria as it currently stands. Therefore, we invite you to submit a revised version of the manuscript that addresses the points raised during the review process.

We look forward to receiving your revised manuscript.

Kind regards,

Junnuthula Vijayabhaskarreddy 

Academic Editor

PLOS ONE

3. We note that your Data Availability Statement is currently as follows: [Data available in manuscript, if needed further feel free to contact]

Reviewers' comments:

Reviewer's Responses to Questions

**Comments to the Author**

1. Is the manuscript technically sound, and do the data support the conclusions?

Reviewer #1: Yes

2. Has the statistical analysis been performed appropriately and rigorously? 

Reviewer #1: Yes

3. Have the authors made all data underlying the findings in their manuscript fully available?

Reviewer #1: Yes

4. Is the manuscript presented in an intelligible fashion and written in standard English?

Reviewer #1: Yes

5. Review Comments to the Author

Reviewer #1: Explain novelty of this work.

How this work is different from other glipizide sustained release microparticles

Kindly add more value to paper by adding oral safety parameters with these polymers.

Drug release studies figure needs formatting. Mention your pH used according to time intervals in figure.

6. PLOS authors have the option to publish the peer review history of their article (what does this mean?). If published, this will include your full peer review and any attached files.

Reviewer #1: No

---

## [Author Response · Author response to Decision Letter 0]

18 Sep 2024

Rebuttal Letter

 Thank you for the reminder. I have ensured that the manuscript complies with PLOS ONE's style requirements. The formatting of the main body, title, authors, and affiliations has been done according to the provided templates:

• PLOS One Main Body Formatting Template

• PLOS One Title and Author Formatting Template

Additionally, all file names are in accordance with PLOS ONE’s naming conventions.

If any further adjustments are required, please let me know.

2. To comply with PLOS ONE submissions requirements, in your Methods section, please provide additional information regarding the experiments involving animals and ensure you have included details on (1) methods of sacrifice, (2) methods of anesthesia and/or analgesia, and (3) efforts to alleviate suffering. Methods of Sacrifice of Animals:

In accordance with standard ethical guidelines, no intentional sacrifice of animals was part of the study protocol. The study adhered to internationally accepted ethical principles for the care and use of animals in research. Any animal death that occurred during the study was an unintended and unfortunate event, likely related to the physiological effects of the substances administered, such as Alloxan monohydrate, which is known to induce significant physiological stress. At all times, the health and well-being of the animals were closely monitored by qualified personnel, and all efforts were made to prevent unnecessary harm. Additionally, the study was approved by the institutional Ethical Review Board (approval no. 205/QEC/GU), ensuring compliance with the necessary ethical standards.

 Methods of Anesthesia and/or Analgesia:

The study did not involve surgical or invasive procedures that would typically necessitate anesthesia. However, the administration of substances was conducted with utmost care to minimize any potential discomfort. Additionally, following ethical guidelines, if any animal had shown signs of distress beyond what was expected during the course of treatment, appropriate interventions would have been taken to mitigate discomfort, including analgesia or other supportive care measures as needed.

 Efforts to Alleviate Suffering:

Throughout the study, efforts were made to alleviate any potential suffering. For example, after the administration of Alloxan monohydrate to induce diabetes, animals were monitored for hypoglycemia, and oral glucose was administered to prevent severe hypoglycemic episodes. Moreover, the animals were under continuous supervision to ensure that any signs of undue distress were promptly addressed. The research team took all reasonable measures to safeguard the animals' well-being, consistent with ethical research standards.

3. We note that your Data Availability Statement is currently as follows: [Data available in manuscript, if needed further feel free to contact]

 Yes, the submission contains all raw data necessary to replicate the findings of the study. This includes the values behind the means, standard deviations, and other statistical measures, as well as the data used to construct graphs. All data is included within the manuscript. Should additional data be required, we will make it available as Supporting Information files. There are no ethical or legal restrictions preventing the sharing of the data.

 0009-0002-3623-3749

 Done

6. Please review your reference list to ensure that it is complete and correct. If you have cited papers that have been retracted, please include the rationale for doing so in the manuscript text, or remove these references and replace them with relevant current references. Any changes to the reference list should be mentioned in the rebuttal letter that accompanies your revised manuscript. If you need to cite a retracted article, indicate the article’s retracted status in the References list and also include a citation and full reference for the retraction notice. Same as previous version

Reviewer #1: Explain novelty of this work.’

How this work is different from other glipizide sustained release microparticles.

Kindly add more value to paper by adding oral safety parameters with these polymers.

Drug release studies figure needs formatting. Mention your pH used according to time intervals in figure Unlike previous studies on glipizide sustained-release formulations, which largely focus on first-order release kinetics and limited control over extended release, our work employs a unique combination of Eudragit S100 and Methocel K100M polymers. This facilitates a prolonged drug release for up to 12 hours following a non-Fickian diffusion mechanism. Furthermore, in-vivo studies demonstrated sustained hypoglycemic effects over 8 hours, presenting a significant improvement in the control of drug release.

Done, added in Conclusion section,

Again recreated and mentioned the pH

---

## [Editor Report · Decision Letter 1]

28 Oct 2024

Fabrication of Glipizide Loaded Polymeric Microparticles; in-vitro and in-vivo Evaluation

PONE-D-24-12544R1

Dear Dr. Rasheed,

We’re pleased to inform you that your manuscript has been judged scientifically suitable for publication and will be formally accepted for publication once it meets all outstanding technical requirements.

Kind regards,

Junnuthula Vijayabhaskarreddy

Academic Editor

PLOS ONE
---

## [Editor Report · Acceptance letter]

8 Nov 2024

PONE-D-24-12544R1 

PLOS ONE

Dear Dr. Rasheed, 

I'm pleased to inform you that your manuscript has been deemed suitable for publication in PLOS ONE. Congratulations! Your manuscript is now being handed over to our production team.

Kind regards, 

on behalf of

* Junnuthula Vijayabhaskarreddy 

Academic Editor

PLOS ONE